# The Removal of Pertechnetate from Aqueous Solution by Synthetic Hydroxyapatite: The Role of Reduction Reagents and Organic Ligands

**DOI:** 10.3390/ijerph20043227

**Published:** 2023-02-12

**Authors:** Oľga Rosskopfová, Eva Viglašová, Michal Galamboš, Martin Daňo, Darina Tóthová

**Affiliations:** 1Department of Nuclear Chemistry, Faculty of Natural Sciences, Comenius University in Bratislava, Mlynská dolina, Ilkovičová 6, 842 15 Bratislava, Slovakia; 2Department of Nuclear Chemistry, Faculty of Nuclear Sciences and Physical Engineering, Czech Technical University in Prague, Břehová 7, 115 19 Prague, Czech Republic

**Keywords:** sorption, hydroxyapatite, technetium-99, organic ligands, radioactive waste

## Abstract

The use of knowledge from technetium radiochemistry (even from nuclear medicine applications) allows us to select an sorbent for ^99m^Tc radionuclide sorption, which is hydroxyapatite. Using radioisotope indication, the ^99m^TcO₄^−^ sorption process on synthetic hydroxyapatite was studied by the batch method in the presence of SnCl_2_ and FeSO_4_ reducing agents. The complexing organic ligands’ effect on the ^99m^TcO₄^−^ sorption under reducing conditions was investigated. In the presence of Sn^2+^ ions without the addition of organic ligand, the sorption percentage reached above 90% independently of the environment. In the presence of Fe^2+^ ions without the addition of organic ligand, the sorption of ^99m^TcO₄^−^ was significantly lower and was at approximately 6%, depending on the concentration of Fe^2+^ ions in solution. The effect of complexing organic ligands on the ^99m^TcO₄^−^ sorption on hydroxyapatite from the aqueous solution, acetate buffer and phosphate buffer decreases in the following order for Sn^2+^: oxalic acid > ethylenediaminetetraacetic acid > ascorbic acid. In the presence of Fe^2+^ ions without organic ligands, the sorption reached up to 15% depending on the composition of the solution. The addition of oxalic acid and ascorbic acid increased the sorption up to 80%. The ethylenediaminetetraacetic acid had no significant effect on the sorption of technetium on hydroxyapatite.

## 1. Introduction

Environmental contamination caused by human activities has an adverse effect on the human body. The nuclear and chemical industries produce hazardous waste that needs to be disposed of. Current technologies for the disposal of waste from aqueous solutions include evaporation, ultrafiltration, precipitation, and ion exchange. The costs of these methods are relatively high. An alternative strategy is to use low-cost natural and synthetic materials as suitable sorbents for heavy metal and radionuclides removal. The selection of an appropriate material is made based on its sorption capacity to immobilize the pollutant and prevent its transport into the wider environment. Radionuclides can occur in the environment in several physico-chemical forms, differing in size, nature of charge and valence, oxidation state, structure, morphology, density, and degree of complexation. Low molecular weight forms can be mobile and potentially bioavailable, while high molecular weight forms can form colloids, polymers, and pseudo-colloids, and are considered as immobile particles [1].

^99^Tc is a long-lived radionuclide (T_1/2_ = 2.11 × 10^5^ y), a fission product with a relatively high fission yield by thermal neutrons of ^235^U and ^239^Pu (6.1%). For this reason, it is one of the radionuclides whose activity in disposed radioactive waste is high even after 10,000 years, and thus represents a potential hazard to the environment because it may be accidently released from the repository [1,2]. The sorption of technetium to the components of the multibarrier system as well as to the host geological formations plays an important role. The nuclear fuel cycle is the predominant source of ^99^Tc in the environment. Power plant accidents, nuclear weapons testing fallout, nuclear power production, the ^99^Tc institutional use, and the ^99m^Tc radiopharmaceutical use (decaying to ground state if ^99^Tc), are much less important sources. Considering the kind of sources and the aqueous chemistry, Tc is released into the environment as pertechnetate anion TcO₄^−^. Given the nature of non-complexing, high water solubility, high volatility, and negative charge of pertechnetate, all of these properties take responsibility for its high mobility in water, making it a problematic nuclide during environmental restoration activities [2]. Based on the facts mentioned above, the Tc behavior in the environment has attracted more and more attention [2,3,4,5,6,7,8]. Considerable effort has been made to understand the long-term biogeochemical behavior of Tc, its transfer in food chains, and the mechanism controlling its mobility in diverse environments [3,7,8,9].

As was mentioned above, low-cost natural and synthetic sorbents could be used as suitable and effective alternative methods for radionuclides removal in order to prevent their spreading into the environment [1], i.e., apatite (hydroxyapatite). Apatite is the general name for a number of phosphate lime minerals. Hydroxyapatite (HA), Ca_10_(PO_4_)_6_(OH)_2_ is the main constituent of phosphate rocks. Natural phosphate rocks contain about 60–70% HA, depending on the location. It occurs naturally as a single mineral or as an isomorphous solid solution. Hydroxyapatite is also an important inorganic component of some biomaterials such as milk, bone, and teeth. Synthetic HA is used for chromatographic and biomedical purposes [10]. Hydroxyapatite has an iso- and hetero-ion exchange ability for both cations and anions, and the ions can be isomorphous, as with Mg^2+^, Sr^2+^, Ba^2+^, Mn^2+^, Zn^2+^, Cd^2+^, Pb^2+^, Eu^3+^, F^−^, Cl^−^, I^−^, AsO_4_^3−^, and VO_4_^3−^, as well as non-isomorphous, such as with M^+^, MoO_2_^2+^, M^3+^, CO_3_^2−^, and SiO^4−^. In general, HA is a suitable sorbent for heavy metals and radionuclides due to its low water solubility, high stability under reducing and oxidizing conditions, high specific surface area, and good buffering properties [11,12,13,14]. These factors are significant because of the potential for immobilization of toxic elements by HA as an important component of soils, bones, and milk. Zhang et al. [15] focused on the removal performance of the three HA powders towards Cd(II) under a series of environmental conditions such as contact time, ionic strength, coexisting inorganic electrolyte ions and organic components, initial Cd(II) concentration, and competing heavy metal ions. The experimental findings demonstrated the significant application potential of Ca-deficient HA material for the cost-effective treatment of Cd(II)-polluted water systems. Hamada et al. [16] investigated the effectiveness of Pb immobilization by hydroxyapatite. Their conclusions were that hydroxyapatite application for Pb immobilization is suitable for soil with a low phosphorus sorption ability. The effects of pH, contact time, fluoride-ion concentration, and the dose of sorbent on the sorption of fluoride ions by hydroxyapatite were studied by Jiménes-Reyes et al. [17]. According to their results, hydroxyapatite is a potential material that can be used for the treatment of water contaminated with fluoride ions.

On the other hand, there is the possibility of using HA, as a natural raw material, to fix radionuclides from polluted water and in nuclear waste technologies, especially those that form large complex ions in the aqueous environment [18,19,20,21,22]. Baybas et al. [23] investigated several types of hydroxyapatite toward the UO_2_^2+^ and Th^4+^ removal. The adsorption dependence on pH and ionic strength provided supportive evidence for the effect of complex formation on the adsorption process. Several other parameters were investigated, as, HA should be considered amongst the favorite adsorbents, particularly for the deposition of nuclear waste containing U and Th, and radionuclide at secular equilibrium with these elements. However, the investigation of UO_2_^2+^ removal by HA was the aim of several newly published papers, with the result being the high HA efficiency toward uranium removal [24,25]. Venkatesan et al. [26] investigated the sorption and immobilization of the several radioactive ionic-corrosion-products i.e., Co^2+^, Cr^3+^, Mn^2+^, Fe^2+^, Ni^2+^, Cu^2+^ and Zn^2+^ generated in nuclear reactor coolant by using a magnetic hydroxyapatite nanocomposite. The samples of HA introduced there were highly able to adsorb and immobilize radioactive waste.

A material for the elimination of radioactive anionic contaminants poses a great challenge. The main aim of the presented work is to investigate the effect of organic ligands on the technetium sorption by hydroxyapatite in the presence of a reducing agent.

## 2. Materials and Methods

### 2.1. Reagents

All chemicals used in experiments were analytical reagent grade pure materials from the chemical supply company Slavus, s.r.o. (Bratislava, Slovak Republic). The synthetic hydroxyapatite with high purity was supplied by the company Sigma-Aldrich (Darmstadt, Germany). ^99m^TcO₄^−^ from ^99^Mo/^99m^Tc generator DRYTEC^TM^ (2.5–100 GBq), GE Healthcare,(Amsterdam, The Nederland) @ 12.00 GMT was used as a radioindicator for sorption process investigation.

### 2.2. Characterization of Hydroxyapatite

The morphology and particle size of the hydroxyapatite sample were examined at 15 kV using a JXA-840A scanning electron microscope (SEM) (Jeol, Ltd., Tokyo, Japan) and a JEM 2000FX transmission electron microscope (TEM) (Jeol, Ltd., Tokyo, Japan) at an accelerating voltage of 160 kV in conjunction with an energy-dispersive (EDS) detector. The EDX Si(Li) detector (Jeol, Ltd., Tokyo, Japan) response function was obtained with a resolution of 130 eV coupled to a JXA-840A SEM and applied to HA chemical composition analysis.

### 2.3. Sorption Experiments

The sorption of TcO₄^−^ on synthetic HA under reducing conditions was studied by the radioisotope indication method using the radioisotope of ^99m^Tc (γ-radiation energy 0.1427 keV). The batch method was used in the static arrangement of the experiment under aerobic conditions at laboratory temperature. Stannous chloride SnCl_2_ and ferric sulphate FeSO_4_ in 0.1 mol·dm^−3^ HCl were used for the reduction of ^99m^TcO₄^−^, respectively. The aqueous phase consisted of distilled water, and acetate buffer (ABS) at pH 4.0, phosphate buffer (PBS) at pH 5.5 and pH 8.0, respectively. The effect of organic ligands on technetium sorption was studied in the presence of oxalic acid (OA), ascorbic acid (AA) and ethylenediaminetetraacetic acid (EDTA).

For the reduction of Tc(VII), 20 mg of SnCl_2_ dissolved in 0.1 mol·dm^−3^ HCl was added to the aqueous phase of ^99m^TcO₄^−^. The concentration of SnCl_2_ in the aqueous phase was equal to 0.02 mol·dm^−3^. The concentration of Fe^2+^ ions ranged from 0.2 to 0.6 mol·dm^−3^. Sorption parameters were determined after adding 5 mL of the aqueous phase to 50 mg of HA in a plastic tube with the tap and solid/liquid phases mixed in a laboratory extractor with a constant speed of mixing. An equilibrium time of 3 h was chosen for the experiments.

After the sorption of ^99m^TcO₄^−^, the suspension was centrifuged for 10 min at 6000 rpm and an supernatant aliquot amount was measured on a 1470 Wizard (PerkinElmer^®^, Sugar Land, TX, USA) gamma counter using an NaI(Tl) detector. The fractional standard deviation of the measurements were below 1%. The redox potential was measured in the system before sorption by using a SenTix^®^ ORP redox electrode (Xylem Analytics GmbH & Co., Weilheim, Germany). The pH of the aqueous phase was measured before and after sorption using an inoLAB pH 720 pH meter (Mettler Toledo, Columbus, OH, USA).

The sorption properties of synthetic hydroxyapatite were calculated by using the following equations:

Distribution coefficient
(1)$$K_{d}=\frac{C_{0}-C_{eq}}{C_{eq}}\;\cdot \frac{V}{m}=\frac{a_{0}-a_{\;}}{a\;}\cdot \frac{V}{m}\;\left({\mathrm{cm}}^{3}\cdot g^{-1}\right)$$

Sorption percentage
(2)$$R=\frac{100\cdot K_{d}}{K_{d}+\frac{V}{m}}\;(\%)$$
where *C*_0_ is initial concentration (mol·dm^−3^), *C_eq_* is the equilibrium concentration (mol·dm^−3^), *V* is the volume of aqueous phase (cm^3^), *m* is the mass of sorbent (g), *a*_0_ is the volume activity of initial solution (Bq·cm^−3^) and *a* is equilibrium volume activity of solution (Bq·cm^−3^).

## 3. Results and Discussion

### 3.1. Characterization of Hydroxyapatite

The SEM images recorded at smaller magnification show that the hydroxyapatite sample consists of agglomerates of tiny particles (Figure 1a). Elements Ca, P and O were detected in these agglomerates using the EDX method, which is in accordance with the calcium phosphate structure (Figure 1b). The results showed that the average content of Ca and P in the synthetic hydroxyapatite sample (obtained by averaging of 10 measurements) were 28.32 at.% and 16.78 at.%, respectively, and the determined Ca/P molar ratio was equal to 1.688. The Ca/P molar ratio value is within a generally accepted range between 1.5 and 1.69 for hydroxyapatite particles. The TEM image showed tiny rod-like nanocrystals creating a sample of hydroxyapatite. Moreover, it is evident that the size of nanocrystals is up to 60 nm in length and up to 15 nm in diameter, which leads to an aspect ratio approaching 4. This was determined by the evaluation of the relevant SAED ring patterns recorded from aggregates of arbitrarily oriented nanocrystals. Sharp rings in both SAED patterns indicate that hydroxyapatite nanocrystals in HA samples exhibit high crystallinity (Figure 1c). The characterization of the studied material was presented in greater detail in our previous work [27].

### 3.2. Sorption Experiments

Hydroxyapatite can bind a variety of cations and anions into its structure. The pH value of the solution is an important parameter that controls the sorption process, since the ionization of functional groups on the HA surface occurs and changes the composition of the solution. The buffering capacity of HA is the result of the acid-base reactions of the available chemical groups on its surface. The phase composition on the hydroxyapatite surface is constant in solution with the pH value in the range of 4–10 [10]. In general, when cations and anions are added to the solution, the initial pH of the solution is lowered due to the ion-exchange reactions on the hydroxyapatite surface, and the dissolution of hydroxyapatite can occur under certain conditions.

The range of technetium oxidation states can make the chemistry difficult to predict and control. High oxidation-state species can form volatile compounds that are particularly difficult in aspects of the nuclear fuel cycle [28]. The most stable and characteristic oxidation state of Tc in weakly acidic, neutral and basic environments is Tc(VII) occurring in the form of pertechnetate ion TcO₄^−^, which was also proved by Chotowski [29]. Tc(IV) can occur in aqueous solutions as TcO^2+^, TcO(OH)^+^ and TcO(OH)_2_, depending on the pH value. The sorption of the ^99m^TcO₄^−^ reduced form to hydroxyapatite in the presence of ligands with the initial pH of the solution before sorption (pH_i_) being around ~2 was studied (Table 1). When Sn^2+^ ions were used for reduction, the pH value after sorption (pH_e_) ranged from 2.4 to 4.2 depending on the amount of complexing ligand added. The redox potential Eh value of the aqueous phase before sorption was in the range of 78–272 mV, corresponding to the Eh value of the reducing environment. The distribution coefficients K_d_ reached values in the order of 10^4^ cm^3^·g^−1^ in the reducing environment without the addition of the complexing ligand in all types of aqueous phases. The addition of AA to the system decreased the K_d_ value to 10^3^–10^2^ cm^3^·g^−1^, and OA and EDTA decreased the K_d_ value to 10^2^–10^1^ cm^3^·g^−1^ depending on the added amount. The studied parameters pH_i_, pH_e_, Eh and K_d_ using the reducing agent SnCl_2_ are shown in Table 1.

In the case of using Fe^2+^ ions for reduction with a concentration of 0.2–0.6 mol·dm^−3^ in solution, the pH after sorption (pH_e_) ranged from 2.1 to 3.5 depending on the amount of complexing ligand added. The Eh value of the redox potential of the aqueous phase before sorption was in the range of 137–289 mV. The distribution coefficients K_d_ reached low values in the reducing environment, with a maximum of the order of 10^2^ cm^3^·g^−1^. The addition of OA to the system increased the K_d_ value by increasing the concentration of Fe^2+^ ions in the aqueous and ABS solution environments, while it did not change significantly in the PBS. The presence of OA in PBS solution significantly affected the values of the distribution coefficient, and the value was in the order of 10^3^ cm^3^·g^−1^. In the case of EDTA, the K_d_ values were very low in all systems. The parameters pH_i_, pH_e_, Eh and K_d_ using the reducing agent FeSO_4_ are shown in Table 2.

In the results obtained from the batch experiments, it was found that the percentage of Tc(IV) sorption (after Tc(VII) reduction by SnCl_2_) onto hydroxyapatite was >99%. The value of the sorption percentage dropped to about 90% after the addition of AA and to 80% with the addition of OA in all studied systems (Figure 2a). In the case of reduction by Fe^2+^ ions, the sorption percentage of Tc to HA without the addition of ligand and after the addition of EDTA was <20%; after the addition of OA, the sorption percentage was >80% in all studied systems (Figure 1b). That can be explained by the formation of an insoluble Tc(IV) compound with oxalates. The presence of AA in water and ABS did not affect the sorption of Tc on HA, while in PBS the sorption percentage increased to more than 87%. Technetium can be reduced without the addition of reducing agent in the presence of ligands to form soluble Tc(IV/V) complexes in alkaline media. Hydroxyl groups play a crucial role in the formation of soluble Tc(V) complexes. EDTA and oxalates are among the ligands with weak reactivity and reduced complex yield to <15% [11]. Tc(IV) forms complexes with phosphates at pH > 5 and sorption on the HA surface is affected by their presence in solution. The sorption of Tc(IV) complexes occurs by ion exchange for phosphate anions, which is due to their higher affinity to Ca sites on the hydroxyapatite surface. Taylor-Pashow et al. in their publications compared SnCl_2_ and FeSO_4_ as reductants for the removal of Tc(VII) from solutions. It was demonstrated that SnCl_2_ was far more effective in removing Tc(VII) than the FeSO_4_ [30,31].

## 4. Conclusions

The sorption of technetium on hydroxyapatite was studied by the batch method using ^99m^TcO₄^−^ as a radioisotope indicator. The effect of complexing organic ligands on technetium sorption under reducing conditions was investigated. The Tc(IV) can occur in aqueous solutions as TcO^2+^, TcO(OH)^+^ and TcO(OH)_2_ depending on the pH value. The pH value after sorption (pH_e_) ranged from 2.4 to 4.2 depending on the amount of complexing ligand added when the Sn^2+^ ions were used for reduction. In the presence of Sn^2+^ ions without the addition of the organic ligand, the sorption percentage reached >90% independent of the aqueous phase. The sorption of technetium to hydroxyapatite after reduction with SnCl_2_ in the presence of organic ligands decreased in the order: oxalic acid > ethylenediaminetetraacetic acid > ascorbic acid. In the case of using Fe^2+^ ions for reduction, the pH after sorption (pH_e_) ranged from 2.1 to 3.5 depending on the amount of complexing ligand added. In the presence of Fe^2+^ reduction ions without organic ligands, the sorption of technetium was <15% depending on the composition of the aqueous phase. After the addition of ascorbic acid and oxalic acid, the sorption increased by up to 80%. Technetium can be reduced without the addition of reducing agent in the presence of ligands to form soluble Tc(IV/V) complexes in alkaline media. Hydroxyl groups play a crucial role in the formation of soluble Tc(V) complexes. The experimental results showed that ^99m^Tc-labeled hydroxyapatite nanoparticles under suitable conditions could be used for the immobilization of ^99m^Tc into hydroxyapatite from liquid radioactive wastes, and in nuclear medicine for targeted cancer treatment in vivo as well.

## Figures and Tables

**Figure 1 ijerph-20-03227-f001:**
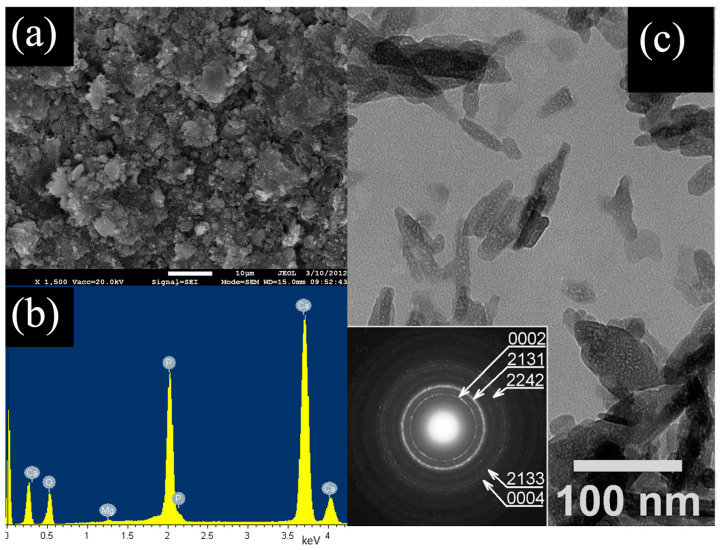
(**a**) SEM micrographs of hydroxyapatite HA, (**b**) EDX spectrum of aggregate consisting of hydroxyapatite, (**c**) TEM image of aggregate of hydroxyapatite nanorods in HAwit SAED ring pattern.

**Figure 2 ijerph-20-03227-f002:**
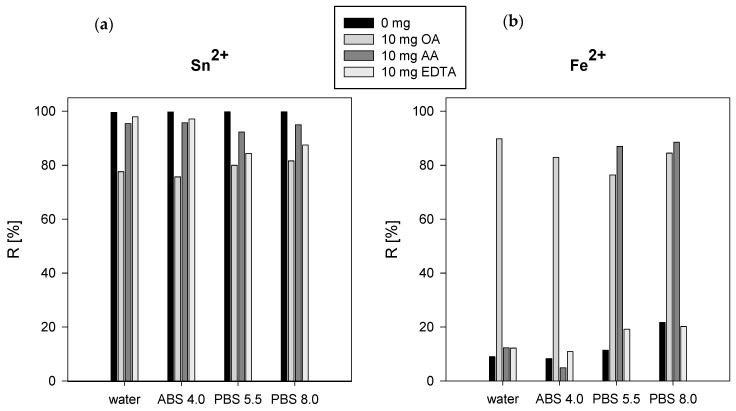
Percentage of ^99m^TcO₄^−^ sorption onto hydroxyapatite after reduction with SnCl_2_ (**a**) and FeSO_4_ (**b**) in the presence of complexing ligands.

**Table 1 ijerph-20-03227-t001:** Parameters of technetium sorption on hydroxyapatite in a reducing environment in the presence of SnCl_2_.

	**m (mg)**	**H_2_O**				**ABS pH 4.0**			
		**pH_i_**	**Eh (mV)**	**pH_e_**	**K_d_ (cm^3^·g^−1^)**	**pH_i_**	**Eh (mV)**	**pH_e_**	**K_d_ (cm^3^·g^−1^)**
	0	2.0	134	2.5	2 × 10^4^	2.0	135	2.7	3 × 10^4^
OA	10	1.8	87	2.9	3 × 10^2^	1.8	93	2.9	3 × 10^2^
	30	1.7	78	2.4	4 × 10^1^	1.7	88	2.4	4 × 10^1^
AA	10	1.9	129	3.0	2 × 10^3^	2.0	125	3.0	2 × 10^3^
	30	1.7	116	3.1	1 × 10^3^	1.7	121	2.8	1 × 10^3^
EDTA	10	1.9	222	2.4	4 × 10^3^	1.9	230	2.6	3 × 10^3^
	30	1.9	241	4.2	6 × 10^1^	1.9	236	4.1	3 × 10^1^
	**m (mg)**	**PBS pH 5.5**				**PBS pH 8.0**			
		**pH_i_**	**Eh (mV)**	**pH_e_**	**K_d_ (cm^3^** **·g^−1^)**	**pH_i_**	**Eh (mV)**	**pH_e_**	**K_d_ (cm^3^** **·g^−1^)**
	0	2.0	146	3.0	4 × 10^4^	2.0	153	3.1	6 × 10^4^
OA	10	1.9	105	2.8	4 × 10^2^	1.9	94	3.0	4 × 10^2^
	30	1.6	80	2.5	4 × 10^1^	1.6	86	2.6	6 × 10^1^
AA	10	2.0	154	3.2	1 × 10^3^	2.0	170	3.0	2 × 10^3^
	30	1.7	145	3.2	7 × 10^2^	1.7	161	3.2	5 × 10^2^
EDTA	10	1.8	272	2.6	5 × 10^2^	1.8	270	2.8	7 × 10^2^
	30	1.8	263	4.1	8 × 10^1^	1.9	257	4.0	5 × 10^1^

**Table 2 ijerph-20-03227-t002:** Parameters of technetium sorption on hydroxyapatite in a reducing environment in the presence of FeSO_4_.

**m (mg)**		**c(Fe^2+^)** **(mol·dm^−3^)**	**H_2_O**				**ABS pH 4.0**			
			**pH_i_**	**Eh (mV)**	**pH_e_**	**K_d_ (cm^3^·g^−1^)**	**pH_i_**	**Eh (mV)**	**pH_e_**	**K_d_ (cm^3^·g^−1^)**
OA	10	0.6	2.0	200	2.8	876	1.6	191	2.6	483
		0.4	2.1	204	3.0	283	1.7	207	2.9	265
		0.2	2.1	216	3.2	98	1.6	218	2.6	82
AA	10	0.6	2.1	262	3.3	14	1.6	249	3.0	5
		0.4	2.0	267	3.3	12	1.8	266	3.3	6
		0.2	2.1	276	3.5	8	1.5	273	3.0	4
EDTA	10	0.6	2.1	185	2.7	14	2.0	289	2.8	12
		0.4	2.1	137	2.8	2	2.0	146	2.9	15
		0.2	2.1	151	3.1	3	2.0	153	3.2	7
**m (mg)**		**c(Fe^2+^)** **(mol** **·dm^−3^)**	**PBS pH 5.5**				**PBS pH 8.0**			
			**pH_i_**	**Eh (mV)**	**pH_e_**	**K_d_ (** **cm^3^** **·g^−1^)**	**pH_i_**	**Eh (mV)**	**pH_e_**	**K_d_ (** **cm^3^** **·g^−1^)**
OA	10	0.6	1.9	188	2.7	324	1.9	188	2.6	547
		0.4	1.7	200	2.6	224	1.9	198	2.6	527
		0.2	1.8	214	2.8	507	2.0	212	3.0	609
AA	10	0.6	1.9	256	2.6	672	1.9	245	2.8	767
		0.4	1.7	260	2.7	949	1.9	250	2.9	799
		0.2	1.7	268	2.9	920	2.0	257	3.0	649
EDTA	10	0.6	1.5	248	2.1	24	1.8	322	2.6	25
		0.4	1.7	133	2.4	17	1.8	142	2.6	22
		0.2	1.8	146	2.8	11	1.8	133	2.8	11

## Data Availability

Not applicable.

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
