# Peer review of "The Removal of Pertechnetate from Aqueous Solution by Synthetic Hydroxyapatite: The Role of Reduction Reagents and Organic Ligands"

_ijerph, 2023, doi:10.3390/ijerph20043227_

Round 1
Reviewer 1 Report
The authors have written a research article entitled “The removal of pertechnetate from aqueous solution by synthetic hydroxyapatite: the role of reduction reagents and organic ligands”. The manuscript is quite interesting, well framed, and based on the 99mTcO₄⁻ sorption process on synthetic hydroxyapatite was studied by the batch method in the presence of SnCl2 and FeSO4 reducing agents. Furthermore, the results indicated that in the presence of Fe2+ ions without organic ligands, the sorption reached up to 15% and the addition of oxalic acid and ascorbic acid increased the sorption up to 80%. The work reported in this manuscript is very interesting and the authors have described the concept to a smaller extent but the manuscript still needs a few Major corrections before acceptance in the International Journal of Environmental Research and Public Health.
However, the following comments need to be addressed.
Comment 1: Grammatical/typographical error issues are so many there in the manuscript at several places also check superscripts and subscripts errors.
Comment 2: The introduction provided a good, generalized background of the topic. However, I think the authors need to cite and discuss some more relevant recent references in the introduction section to strengthen the section.
Comment 3: Impact and novelty have to be clarified in the introduction section.
Comment 4: Figure 1 scale bar exact size is missing in both SEM and TEM images, so provide the original images of the SEM and TEM images.
Comment 5: Figures 1 (a) & 1(c) is not cited in the manuscript text. So cite Figures 1 (a) & 1(c) at the appropriate place.
Comment 6: SAED pattern is mentioned in the manuscript text. However, no figure is provided in the manuscript. So include the SAED pattern Figure in the revised manuscript.
Comment 7: Line 147, “TEM images” should be “TEM image (Figure 1(c))”.
Comment 8: Overall, the “Discussion” section needs more comparative study than just mentioning the obtained results. So please compare the obtained results with previous studies. Discussion on this point has to be strengthened.
Comment 9: Conclusions: This section is quite too general. Include in this section the most important findings to highlight the importance of this study.
Comment 10: Please add the Graphical abstract figure to explain the outline of this research to attain a broad readership.
Author Response
Dear reviewer, thank you very much for your comments, please you will find the answers to your comments and suggestions below. The manuscript was improved by incorporating your comments. The changes are provided by the function “track changes”. The English was checked by native speaker, and some proves were provided.
Authors

Reviewer 2 Report
Dear authors,
The topic of the work is current and you presented good and interesting results.
A better way to connect hydroxyapatite in the introductory part with the rest of the text
In section 2.2, you stated that you used the BET method, I did not see in the results isotherms, pore distribution and other analyses?
Experiments are missing,
Should the FTIR method be used to see the chemistry of the surface and the functional group? if possible, should the chemical kinetics be determined?
The EDX spectrum should be better explained
In my opinion, abbreviations should be placed in the continuation of the text of the paper so that the readers could read more easily and not at the end of the paper.
Have you tried any other material for the removal of pertechnetate from aqueous solution?
You need to do additional analysis and explain everything and put it together as a whole, if you make a significant change, the work can be accepted.
Author Response

(The authors gave the same response as above.)

Reviewer 3 Report
First I would like to thank the authors for choosing an important point and taking into account the economic and environmental aspects.
There are some important notes that are useful in improving the paper and making it better, such as:
1- I am not found word Figure 1 or 2 within context of the research
2- all abbreviations and symbols on the paper must be completed
Please find attached file
thanks

Author Response

(The authors gave the same response as above.)

Author Response

(The authors gave the same response as above.)

Round 2
Reviewer 1 Report
The authors revised their manuscript according to my comments. So the manuscript is suitable for acceptance.
Reviewer 2 Report
My opinion is that the work should be accepted as is.